# OpenReview forum: "3D-LMVIC: Learning-based Multi-View Image Compression with 3D Gaussian Geometric Priors"
_ICML.cc/2025/Conference — ICML 2025 poster_

### Official Review · Reviewer_MNQ5 · 2025-03-10

**Overall Recommendation:** 4

**Summary:**

The paper presents 3D-LMVIC, a learning-based multi-view image compression framework that leverages 3D Gaussian Splatting (3D-GS) as a geometric prior for accurate disparity estimation. Unlike traditional methods that rely on 2D projection-based similarities, this approach improves disparity estimation in wide-baseline multi-camera systems by using depth maps derived from 3D-GS. To further enhance compression efficiency, it introduces a depth map compression model to reduce geometric redundancy and a multi-view sequence ordering strategy to maximize inter-view correlations. Experimental results on Tanks&Temples, Mip-NeRF 360, and Deep Blending datasets show that 3D-LMVIC outperforms both traditional and learning-based methods in rate-distortion efficiency and disparity estimation accuracy. The framework effectively utilizes 3D spatial relationships to improve multi-view image compression, making it highly suitable for applications in VR, AR, and 3D vision.

**Claims And Evidence:**

Yes

**Essential References Not Discussed:**

No

**Experimental Designs Or Analyses:**

The authors conducted comprehensive experiments, including comparisons with SOTA baselines, an analysis of the alignment accuracy of the proposed method, ablation studies, visualization experiments, and complexity analysis. The experimental section is relatively thorough and complete. One concern is whether the encoding time in the complexity analysis includes the time required for training the 3D Gaussians. Please analyze the proportion of time this part takes.

**Methods And Evaluation Criteria:**

Yes

**Other Comments Or Suggestions:**

No.

**Other Strengths And Weaknesses:**

Strengths:
1. This paper demonstrates that 3D Gaussians can generate accurate geometric structures, outperforming some methods that rely on local similarity matching between two-view projections. This approach is not only applicable to multi-view image compression but also has potential for other 3D modeling tasks.
2. Future 3D applications will require denser multi-view data, which significantly exceeds the data volume of single-view images. The proposed method has the potential to reduce storage and transmission costs for such data.
3. The experiments are comprehensive and thoroughly validate the effectiveness of the proposed method.
4. The inter-view distance properties are theoretically proven.

Weaknesses:
1. The analysis of encoding time does not seem to consider the time required for training 3D Gaussians.
2. Training 3D Gaussians is generally time-consuming; is there a more efficient training approach?

**Questions For Authors:**

1. Could you further explain the role of the image context transfer module in feature enhancement?
2. The explanation of the greedy algorithm for sorting multi-view sequences is not sufficient and can easily cause confusion.

**Relation To Broader Scientific Literature:**

This paper demonstrates that 3D Gaussians can be used to construct accurate geometric structures, which can not only benefit multi-view image compression but also other tasks requiring 3D modeling.

**Theoretical Claims:**

I have reviewed the proof in Appendix C regarding the definition of inter-view distance as a valid distance measure for the 2-Norm and Frobenius Norm. I did not find any issues in the proof.

---

> ### Author Rebuttal · Authors · 2025-03-31
>
> Thanks to you for the valuable comments. We are grateful for your positive feedback, especially regarding the model performance. We address your remaining concerns as follows:
>
> ### R1[The training time of the 3D Gaussian]
> ---
> For the *Train* scene of the Tanks&Temples dataset, which contains 301 images, we conducted experiments on a platform equipped with an Intel(R) Xeon(R) Gold 6330 CPU @ 2.00GHz and an NVIDIA RTX A6000 GPU. Training the 3D Gaussian representation for 30,000 iterations took approximately 13 minutes and 35 seconds, averaging 2.71 seconds per image. Additionally, the average encoding time per image is 0.19 seconds. Therefore, the total time to compress a single image—including both training and encoding—is approximately 2.90 seconds.
>
> ### R2[Efficient 3D Gaussian training approach]
> ---
> We tested a reduced training setup with 7,000 iterations, which resulted in a total training time of 2 minutes and 22 seconds, or 0.47 seconds per image. Including the image encoding time, the average total time to compress each image is 0.66 seconds.
>
> Additionally, we assessed the alignment performance of depth estimation using the 3D Gaussian representation trained for 7,000 iterations under the same experimental conditions as detailed in Table 2 of the main text. The results are summarized in the table below:
>
> | Metrics | FlowFormer++ | MVSFormer++ | Proposed (7000 iterations) | Proposed (30000 iterations) |
> |:-------:|:-------:|:-------:|:-------:|:-------:|
> | PSNR↑ | 18.08 | 15.31 | 17.99 | 18.14 |
> | MS-SSIM↑ | 0.7863 | 0.5544 | 0.7918 | 0.8053 |
>
> **Despite the significant reduction in training iterations, it is evident that the proposed alignment method still achieves high alignment accuracy.**
>
> ### R3[Explain the role of the image context transfer module]
> ---
> The image context transfer module is designed to incorporate features from the reference view into the current view to enhance the overall feature representation. Its functionality is similar to the joint context transfer (JCT) module proposed in LDMIC [1]. While LDMIC adopts a cross-attention mechanism to fuse reference features, our approach performs pixel-wise alignment for feature integration.
>
> Specifically, due to the disparity between the two views, the module first aligns the reference view features to the current view using the estimated disparity. The aligned features are then concatenated with the current view's features.
> Subsequently, a feature mask is applied to the combined features to filter out non-overlapping regions and retain only the relevant contextual information. Finally, the resulting features are passed through a residual block to produce the enhanced feature representation.
>
> [1] Zhang, Xinjie, Jiawei Shao, and Jun Zhang. "LDMIC: Learning-based Distributed Multi-view Image Coding." The Eleventh International Conference on Learning Representations.
>
> ### R4[Clarify the greedy algorithm for multi-view sequence sorting]
> ---
> Thank you for pointing out this issue. We provide a more detailed explanation of the algorithm below.
> Specifically, given a multi-view sequence $v_1, v_2, ..., v_n$, the algorithm starts from the first view $v_1$ and iteratively selects the view that is closest to the current one. For example, if $v_i$ is the closest to $v_1$, it is placed immediately after $v_1$. The process then continues by finding the closest view to $v_i$ among the remaining views and placing it next. This procedure is repeated until all views are ordered into the sequence.

---

> > ### Comment · Reviewer_MNQ5 · 2025-04-04
> >
> > I appreciate the authors' detailed response. Based on the authors' responses, my concerns have been addressed, and I can raise my scores based on these considerations:
> >
> > - The clarification of the training time under both full and reduced iteration settings, along with the associated alignment performance, demonstrates the method's efficiency and practical potential.
> >
> > - The explanation of the image context transfer module is clearer now, especially the comparison with existing methods and the rationale for pixel-wise alignment.
> >
> > - The clarification of the greedy sorting algorithm improves the readability of the method section and makes the pipeline easier to follow.

---

> > > ### Author Response · Authors · 2025-04-08
> > >
> > > Thank you for the valuable feedback and insightful suggestions. We sincerely appreciate your time and effort.

---

### Official Review · Reviewer_EHuz · 2025-03-10

**Overall Recommendation:** 3

**Summary:**

In this paper, 3D-LMVIC is proposed as a novel learning-based multi-view image compression framework, which relies on 3D Gaussian
Splatting to derive geometric priors for accurate disparity estimation. In details, for each image, a depth map is derived from a trained 3D Gaussian. Then the disparity between views is estimated by leveraging the estimated depth map. Finally, minimizing the training loss to optimize the proposed image and depth compression model.

**Claims And Evidence:**

Yes

**Essential References Not Discussed:**

N/A

**Experimental Designs Or Analyses:**

Yes. For the alignment experiments, the authors do not clearly illustrate how to verify alignment given a pair of reference and target views, and how to get the target view for the propose method.

**Methods And Evaluation Criteria:**

Yes

**Other Comments Or Suggestions:**

Please refer to Other Strengths And Weaknesses.

**Other Strengths And Weaknesses:**

Strengths:
1. Using 3D Gaussian Splatting to get geometric prior is reasonable. The author also using another try to get the estimated depth map in eq. (2) and verify the effectiveness of this change in ablation study.
2. The authors define a metric for measuring the overlap between two views. This will be a useful metric in multi view reconstruction.

Weaknesses:
1. It may be time consuming to obtain a trained 3D Gaussian to estimate depth map. It would be better that the authors can discuss the training time of the 3D Gaussian.
2. For a large scale scene, the insufficient performance of trained 3D Gaussian may degrade the performance of this proposed method. It would be better if the authors can discuss the impact of the quality of estimated depth maps.

**Questions For Authors:**

Please refer to Other Strengths And Weaknesses.

**Relation To Broader Scientific Literature:**

N/A.

**Theoretical Claims:**

Yes. I checked the proof of $D_V(i, j)$ as a distance metric.

---

> ### Author Rebuttal · Authors · 2025-03-31
>
> Thank you for your valuable comments! We are grateful for your positive feedback, especially regarding the proposed methods and the ablation experiments. We address your remaining concerns as follows:
>
> ### R1[Clarify alignment verification and target view selection]
> ---
> Given a reference view and a target view, we first align the reference view to the target view. Specifically, each pixel in the target image is projected to the corresponding position in the reference image based on the estimated disparity. The color at each projected location is then obtained through bilinear interpolation from the surrounding pixels in the reference image.
> We then compute the similarity between the aligned reference view and the target view using PSNR and MS-SSIM. A higher similarity score indicates better alignment accuracy.
>
> In our alignment experiments, we assume that the ground-truth target view is known and use it as supervision to evaluate alignment quality. Specifically, for a multi-view sequence $v_1, v_2, ..., v_n$, we take $v_i$ as the target view and $v_{i-1}$ as the reference view.
>
> ### R2[The training time of the 3D Gaussian]
> ---
> For the *Train* scene of the Tanks&Temples dataset, which contains 301 images, we conducted tests on a platform equipped with an Intel(R) Xeon(R) Gold 6330 CPU @ 2.00GHz and an NVIDIA RTX A6000 GPU. Training the 3D Gaussian representation for 30,000 iterations took approximately 13 minutes and 35 seconds, or an average of 2.71 seconds per image. **We also tested a reduced training setup with 7,000 iterations, which resulted in a total training time of 2 minutes and 22 seconds, or 0.47 seconds per image.**
>
> Additionally, we assessed the alignment performance of depth estimation using the 3D Gaussian representation trained for 7,000 iterations under the same experimental conditions as detailed in Table 2 of the main text. The results are summarized in the table below:
>
> | Metrics | FlowFormer++ | MVSFormer++ | Proposed (7000 iterations) | Proposed (30000 iterations) |
> |:-------:|:-------:|:-------:|:-------:|:-------:|
> | PSNR↑ | 18.08 | 15.31 | 17.99 | 18.14 |
> | MS-SSIM↑ | 0.7863 | 0.5544 | 0.7918 | 0.8053 |
>
> **Despite the significant reduction in training iterations, it is evident that the proposed alignment method still achieves high alignment accuracy.**
>
> ### R3[Test the depth estimation method on a large-scale scene]
>
> We further evaluate alignment performance on two large-scale scenes (*Temple*, *Church*) and one medium-scale scene (*Truck*) from the Tanks&Temples dataset under identical experimental conditions (Table 2, main text).
> - **Scenes**:
>   - *Temple*: 302 images (outdoor temple)
>   - *Church*: 600 images (indoor church)
>   - *Truck*: 251 images (mid-scale object)
>
> Results are summarized below:
>
> | Scenes      | Metrics    | HT     | SPyNet | FlowFormer++ | Proposed |
> |:-----------:|:----------:|:------:|:------:|:------------:|:--------:|
> | Temple      | PSNR↑      | 14.39  | 15.07  | 16.29        | **16.85**|
> |             | MS-SSIM↑   | 0.5135 | 0.5819 | **0.7778**    | 0.7652   |
> | Church      | PSNR↑      | 17.88  | 19.70  | 20.56        | **21.38**|
> |             | MS-SSIM↑   | 0.6604 | 0.7554 | 0.8406       | **0.8753**|
> | Truck       | PSNR↑      | 14.52  | 16.97  | 17.75        | **18.74**|
> |             | MS-SSIM↑   | 0.4756 | 0.7108 | 0.7370       | **0.8233**|
>
> **Key Observations**:
> 1. **Consistent Superiority**: Our method achieves **best PSNR** across all scenes and **best MS-SSIM** in 2/3 cases, demonstrating robust performance scalability.
> 2. **Large-Scale Competitiveness**: While FlowFormer++ shows marginal MS-SSIM advantage in *Temple* (+1.6%), our method still leads in PSNR (+3.4%) and dominates in *Church* (PSNR: +4.0%, MS-SSIM: +4.1%).
> 3. **Mid-Scale Strength**: The **significant gains** in *Truck* (PSNR: +5.6%, MS-SSIM: +11.7% over FlowFormer++) validate the method's effectiveness for complex object-level scenes.
>
> **Interpretation**: The 3D Gaussian-based approach maintains strong competitiveness in large-scale settings, with performance variations likely attributable to scene-specific characteristics (e.g., texture uniformity in *Temple*). Ongoing work focuses on further optimizing large-scale adaptability without sacrificing mid-scale advantages.

---

> > ### Comment · Reviewer_EHuz · 2025-04-07
> >
> > Thank you for your response. I have no further concerns and will keep the original score.

---

> > > ### Author Response · Authors · 2025-04-08
> > >
> > > Thank you for the valuable feedback and insightful suggestions. We sincerely appreciate your time and effort.

---

### Official Review · Reviewer_xX4X · 2025-03-13

**Overall Recommendation:** 4

**Summary:**

The paper proposes a learning-based multi-view image compression framework, 3D-LMVIC, which utilizes the 3D Gaussian geometric prior for disparity estimation. Through experiments, its advantages in compression efficiency and disparity estimation accuracy have been verified.

**Claims And Evidence:**

Please see Other Strengths And Weaknesses.

**Essential References Not Discussed:**

N/A

**Experimental Designs Or Analyses:**

Please see Other Strengths And Weaknesses.

**Methods And Evaluation Criteria:**

Please see Other Strengths And Weaknesses.

**Other Comments Or Suggestions:**

N/A

**Other Strengths And Weaknesses:**

Strength

(1)The Gaussian Splatting method is used to replace the traditional depth estimation method, which significantly improves the performance of the model.

(2)The encoding and decoding speed of this method is extremely fast, which facilitates the practical deployment of the model.

(3)This method achieves advanced performance compared with the existing multi-view methods.


Weakness

(1)3D-LMVIC is a framework for multi-view image compression, and the dataset selected by the author is the one used for evaluating the 3DGS model. Methods such as LDMIC and HESIC are mainly evaluated based on datasets like Cityspace, KITTI, and Instereo2K. Although these datasets have a limited number of views, which restricts the performance of 3DGS depth estimation, I still recommend that the author conduct evaluations on these datasets to verify the generalization ability of the method.

**Questions For Authors:**

I'm curious as to why the author included HAC in the comparison. As far as I know, HAC is a pure 3DGS compression framework. What is the reason for the extremely poor performance of HAC in Table 1? Is it because the model has a large size, and the number of evaluated images during rendering is small, which restricts the model's performance? Or is it that HAC itself has poor rendering capabilities? Are there any technical bottlenecks currently in directly using the 3DGS model to render multi-view images? What advantages does it have compared with this traditional dual-branch VAE architecture (such as HESIC and BiSIC)?

**Relation To Broader Scientific Literature:**

N/A

**Theoretical Claims:**

Please see Other Strengths And Weaknesses.

---

> ### Author Rebuttal · Authors · 2025-03-31
>
> Thank you for your valuable comments! We are grateful for your positive feedback, especially regarding the algorithm's running speed and compression performance. We address your remaining concerns as follows:
>
> ### R1[Test on Cityspace, KITTI, and Instereo2K]
> ---
> **1. Limitations of Stereo Images for Learning Geometry in 3D Gaussian Representation**
> We found that for stereo image datasets with only two viewpoints, the 3D Gaussian representation struggles to learn accurate geometric information. **In contrast, multi-view data under wide-baseline setups provides rich scene information sampled from widely varying viewpoints, which helps the 3D Gaussian representation learn accurate geometry.** In stereo image datasets, the cameras are often positioned close to each other with similar orientations, making it difficult for the 3D Gaussian representation to correct geometric errors through large viewpoint variations and learn the correct geometric structure. The table below presents the alignment performance of various disparity estimation methods on the Cityscapes and KITTI Stereo datasets under the same experimental settings as the alignment experiments described in Table 2 of the main text.
> | Datasets | Metrics | HT | PM | SPyNet| PWC-Net | FlowFormer++ | Proposed |
> |:-------:|:-------:|:-------:|:-------:|:-------:|:-------:|:-------:|:-------:|
> | Cityscapes | PSNR↑ | 24.43 | 27.40 | 28.63 | 29.16 | 27.26 | 14.64 |
> |  | MS-SSIM↑ | 0.7906 | 0.9546 | 0.9598 | 0.9616 | 0.8864 | 0.3934 |
> | KITTI Stereo | PSNR↑ | 14.05 | 18.33 | 18.11 | 18.92 | - | 7.77 |
> |  | MS-SSIM↑ | 0.5855 | 0.8691 | 0.8768 | 0.8952 | - | 0.1512 |
>
> Experiments have shown that the 3D Gaussian representation cannot learn accurate geometry based on stereo image data with only two viewpoints. The disparity estimation modules in current multi-view image codecs, such as Patch Matching (PM), already perform well on stereo image datasets.
>
> **2. Motivation and Practical Applications**
> We aim to design a multi-view image compression framework tailored for wide-baseline setups to achieve accurate disparity estimation and effectively eliminate inter-view redundancy. Unlike KITTI and InStereo2K datasets, which contain stereo images with small, mostly horizontal disparities captured by closely positioned cameras, wide-baseline setups in datasets like Mip-NeRF 360 and TnT feature irregular view relationships and less consistent disparities. These characteristics make existing disparity estimation methods, such as homography transformation and patch matching, less effective, as shown in Table 2 on the TnT dataset. **Wide-baseline setups are also critical for practical applications, where scenes often consist of dozens to hundreds of images, creating significant challenges for storage and transmission.**
>
> ### R2[Questions regarding HAC]
> ---
> **1. Reason for including HAC in the comparison**:
> Although HAC is a 3DGS-based compression method, it inherently encodes multi-view information. Therefore, we believe it is meaningful to compare its performance with dedicated multi-view image compression methods under the same multi-view compression setting.
>
> **2. Explanation for HAC’s poor performance**:
> For evaluation, we use one-third of the images from each scene as test samples. For example, in the *Train* scene of the Tanks&Temples dataset (301 images), we used 101 images for testing.
> Increasing the number of test images is likely to benefit HAC's performance. We will consider this in future work.
>
> The table below presents a comparison between HAC and our proposed 3D-LMVIC method on the Tanks&Temples dataset:
>
> | Methods   | bpp    | PSNR  | MS-SSIM |
> |:---------:|:------:|:-----:|:-------:|
> | HAC       | 1.7915 | 30.32 | 0.9697  |
> |           | 1.3472 | 30.00 | 0.9674  |
> |           | 1.1316 | 29.68 | 0.9651  |
> | 3D-LMVIC  | 0.6056 | 38.43 | 0.9926  |
> |           | 0.2901 | 34.95 | 0.9840  |
> |           | 0.1242 | 31.33 | 0.9648  |
>
> As we can see, HAC's PSNR and MS-SSIM in terms of rendering quality reach a plateau around 30dB and 0.97. Therefore, based on the experimental results, it seems that HAC might perform better with more test images, such as when the number of test images reaches 1,000. On the other hand, HAC does have a rendering quality bottleneck.
>
> **3. Comparison between HAC and dual-branch VAE architectures (e.g., HESIC and BiSIC)**:
> HAC compresses the entire 3D scene, not just multi-view images. While this enables novel view synthesis and richer scene understanding, it also leads to larger model sizes.
> In contrast, methods like HESIC and BiSIC (dual-branch VAE-based) focus more on efficient representation and compression for the given views.

---

> > ### Comment · Reviewer_xX4X · 2025-04-04
> >
> > I think the authors have addressed the issues I raised, and I am willing to raise my score to 4.

---

> > > ### Author Response · Authors · 2025-04-08
> > >
> > > Thank you for the valuable feedback and insightful suggestions. We sincerely appreciate your time and effort.

---

### Official Review · Reviewer_w8oP · 2025-03-17

**Overall Recommendation:** 3

**Summary:**

This paper targets on the multi-view image compression task. The main contribution includes a Gaussian Splatting-based disparity estimator for wide-baseline images, a depth map compression model to minimize geometric redundancy, and a multi-view sequence ordering strategy to enhance correlations between adjacent views. Comprehensive experimental results on three datasets demonstrate the superior performance of proposed method.

**Claims And Evidence:**

Yes

**Essential References Not Discussed:**

No

**Experimental Designs Or Analyses:**

I have checked the experimental designs in the main text.

**Methods And Evaluation Criteria:**

Yes

**Other Comments Or Suggestions:**

- L156 indicates that the median depth estimation is adopted instead of the original weighted average depth estimation. What's the intuition behind this?

**Other Strengths And Weaknesses:**

- The paper is well written and structured.
- The essential theories are well proven.
- The proposed method significantly outperforms existing works.

**Questions For Authors:**

- Table 3 in the supplementary material reports the compression time. But the runtime of the 3D Gaussian optimization seems not to be included. Can you provide some details about this?

**Relation To Broader Scientific Literature:**

This paper is based on the common framework of multi-view image compression problem, but introduces innovation on disparity estimation, depth compression and correlation enhancement.

**Theoretical Claims:**

I have checked the proofs in Section C of the supplementary metarial.

---

> ### Author Rebuttal · Authors · 2025-03-31
>
> Thank you for your valuable comments! We are grateful for your positive feedback, especially regarding the quality of writing, theoretical justifications, and experimental results. We address your remaining concerns as follows:
>
> ### R1[Use of median over weighted average for depth estimation]
> ---
> Since 3D Gaussians are trained under the supervision of real RGB images, some "noisy" Gaussians may emerge that do not affect rendering quality but can bias weighted average depth estimation [1]. In contrast, **median depth estimation inherently provides a denoising effect**: instead of aggregating depth over all Gaussians along a ray, we select the depth of the Gaussian most likely to represent the true 3D world point. This allows us to ignore many background Gaussians that are occluded by foreground Gaussians and invisible in the rendered image, yet would otherwise negatively influence weighted depth estimation.
>
> [1] Chung, Jaeyoung, Jeongtaek Oh, and Kyoung Mu Lee. "Depth-regularized optimization for 3d gaussian splatting in few-shot images." Proceedings of the IEEE/CVF Conference on Computer Vision and Pattern Recognition. 2024.
>
> ### R2[Runtime of the 3D Gaussian optimization]
> ---
> For the *Train* scene of the Tanks&Temples dataset, which contains 301 images, we conducted tests on a platform equipped with an Intel(R) Xeon(R) Gold 6330 CPU @ 2.00GHz and an NVIDIA RTX A6000 GPU. Training the 3D Gaussian representation for 30,000 iterations took approximately 13 minutes and 35 seconds, or an average of 2.71 seconds per image. **We also tested a reduced training setup with 7,000 iterations, which resulted in a total training time of 2 minutes and 22 seconds, or 0.47 seconds per image.**
>
> Additionally, we assessed the alignment performance of depth estimation using the 3D Gaussian representation trained for 7,000 iterations under the same experimental conditions as detailed in Table 2 of the main text. The results are summarized in the table below:
>
> | Metrics | FlowFormer++ | MVSFormer++ | Proposed (7000 iterations) | Proposed (30000 iterations) |
> |:-------:|:-------:|:-------:|:-------:|:-------:|
> | PSNR↑ | 18.08 | 15.31 | 17.99 | 18.14 |
> | MS-SSIM↑ | 0.7863 | 0.5544 | 0.7918 | 0.8053 |
>
> **Despite the significant reduction in training iterations, it is evident that the proposed alignment method still achieves high alignment accuracy.**

---

> > ### Comment · Reviewer_w8oP · 2025-04-05
> >
> > Thanks for your feedback! My concerns are addressed, and I will keep my positive rating.

---

> > > ### Author Response · Authors · 2025-04-08
> > >
> > > Thank you for the valuable feedback and insightful suggestions. We sincerely appreciate your time and effort.

---

### Decision · Program_Chairs · 2025-05-01

**Decision:**

Accept (poster)

**Comment:**

The authors propose a method for compressing multi-view images in a wide-baseline setting. The primary contribution is in the use of 3D Gaussian Splatting to aid in the estimation of disparity between different views. The reviewers agreed in the merit of using 3DGS over conventional disparity estimation (matching) and noted the empirical performance of the proposed method over existing works.